# Novel Biomarkers for Personalized Cancer Immunotherapy

**DOI:** 10.3390/cancers11091223

**Published:** 2019-08-22

**Authors:** Yoshitaro Shindo, Shoichi Hazama, Ryouichi Tsunedomi, Nobuaki Suzuki, Hiroaki Nagano

**Affiliations:** 1Department of Gastroenterological, Breast and Endocrine Surgery, Yamaguchi University Graduate School of Medicine, Ube 755-8505, Japan; 2Department of Translational Research and Developmental Therapeutics against Cancer, Yamaguchi University Faculty of Medicine, Ube 755-8505, Japan

**Keywords:** cancer immunotherapy, biomarkers, immune checkpoint inhibitors, programmed cell death ligand 1, tumor mutation burden, neoantigen, tumor microenvironment, Ki-67 expression, microbiome

## Abstract

Cancer immunotherapy has emerged as a novel and effective treatment strategy for several types of cancer. Immune checkpoint inhibitors (ICIs) have recently demonstrated impressive clinical benefit in some advanced cancers. Nonetheless, in the majority of patients, the successful use of ICIs is limited by a low response rate, high treatment cost, and treatment-related toxicity. Therefore, it is necessary to identify predictive and prognostic biomarkers to select the patients who are most likely to benefit from, and respond well to, these therapies. In this review, we summarize the evidence for candidate biomarkers of response to cancer immunotherapy.

## 1. Introduction

Cancer immunotherapy has emerged as an effective and promising treatment strategy in addition to surgery, chemotherapy, and radiotherapy, for various cancers. Several immunotherapeutic strategies, including non-specific biological response modifiers [1], interleukin (IL)-2-activated lymphocytes [2], tumor-specific reactive CD8+ T-lymphocyte transfer [3,4], dendritic cell (DC) vaccines [5,6], and tumor-associated antigen (TAA)-derived peptides [7,8], have been used for cancer treatment. Sipuleucel-T, an autologous antigen-specific vaccine, was the first cancer immunotherapy approved by the U.S. Food and Drug Administration (FDA) in 2010 for patients with asymptomatic or minimally symptomatic castration-resistant prostate cancer [9]. The efficacy of these immunotherapies was limited due to the immunosuppressive tumor microenvironment and the selection of patients with late-stage cancer [10,11]. However, immunotherapies targeting in vivo induction of tumor-specific cytotoxic T lymphocytes (CTLs) have demonstrated clinical response in a fraction of patients with advanced cancers [8,12,13]. Therefore, it is important to identify predictive biomarkers that can enable, prior to treatment, the selection of patients who are expected to obtain clinical benefit from immunotherapy.

Recently, immune checkpoint inhibitors (ICIs), including inhibitors of cytotoxic T-lymphocyte antigen-4 (CTLA-4), programmed cell death receptor-1 (PD-1), and its ligand (PD-L1), have demonstrated durable clinical responses in several cancers [14,15,16]. The fundamental mechanism of ICIs involves the induction of effective antitumor immune responses by eliminating the immunosuppressive state of the tumor [17,18]. However, ICI as single agents demonstrated response rates of lower than 30% in patients with various cancers [16,19]. A combination therapy of ICIs can improve the response rate, but is associated with increased immune-related toxicity and treatment costs [20].

In this era of precision medicine, it is necessary to identify predictive and prognostic biomarkers to identify the patients who are most likely to benefit from, and respond well to, cancer immunotherapies. Risk prediction for immune-related adverse events, such as dermatitis, hypothyroidism, colitis, and hepatitis, is also required. In this review, we summarize the evidence for candidate biomarkers of response to cancer immunotherapy, focusing on the tumor tissue, peripheral blood, and other sources (Table 1).

## 2. Biomarkers in Tumors and the Tumor Microenvironment

### 2.1. PD-L1

PD-1 is a key immune checkpoint receptor that is expressed in activated T cells. The binding of PD-1 with its ligand, PD-L1, negatively regulates T cells, causing decreased proliferation and the production of effector cytokines. PD-L1 can also be expressed in tumor cells in various cancers, and contributes to tumor immune evasion [17]. The anti-PD-1 antibody inhibits the PD-1/PD-L1 interaction, which enables tumor-reactive T cells to kill cancer cells. A phase I clinical study of the anti-PD-1 antibody was conducted in 42 patients with melanoma, non-small-cell lung cancer (NSCLC), colorectal cancer (CRC), renal cell cancer (RCC), or prostate cancer. Of these patients, 36% with PD-L1-positive tumors showed an objective response, whereas none of the patients with a PD-L1-negative result showed an objective response [16]. In another phase I trial of pembrolizumab, a monoclonal antibody targeting PD-1, in patients with advanced NSCLC (KEYNOTE-001), increased PD-L1 expression was associated with a better treatment response and longer progression-free survival (PFS). In this trial, PD-L1 expression was determined by the tumor proportion score (TPS) and classified into < 1%, 1‒49%, and ≥50%. The objective response rates (ORRs) and median PFS were 10.7% and 4.0 months in TPS <1% tumors, 16.5% and 4.1 months in TPS 1‒49% tumors, and 45.2% and 6.3 months in TPS ≥50% tumors, respectively [21]. These results indicated that high PD-L1 expression is associated with an increased response rate and clinical benefit for ICIs. Several studies have also demonstrated the potential of PD-L1 expression as a predictive biomarker for response to anti-PD-1/PD-L1 inhibitors [22,23,24,25].

Conversely, it has been reported that patients with low or negative PD-L1 expression in tumors exhibit clinical benefit from anti-PD-1/PD-L1 treatment. In a phase III trial of nivolumab—an anti-PD-1 antibody—metastatic melanoma patients with PD-L1 positive tumors showed an ORR of 52.7% and those with PD-L1 indeterminate or negative tumors demonstrated an ORR of 33.1% [68]. In another phase II trial of nivolumab, 18% of metastatic RCC patients with low or negative PD-L1 expression (<5% on tumor cells) and 31% of those with positive PD-L1 expression (≥5% on tumor cells) responded to treatment [69]. Moreover, based on a ≥1% cutoff for PD-L1 expression, ORRs in PD-L1-positive and PD-L1-negative patients were similar. Although high PD-L1 expression is associated with a high response rate, it cannot be used as a predictive biomarker for the selection or exclusion of patients treated with anti-PD-1/PD-L1 inhibitors. In addition, other predictive biomarkers of response to anti-PD-1/PD-L1 treatment might be involved.

Although PD-L1 expression in tumors can be assessed by immunohistochemistry (IHC), there are several limitations to using PD-L1 expression as a biomarker. Firstly, there is spatial and temporal heterogeneity of PD-L1 expression within the tumor; moreover, PD-L1 expression is affected by prior therapies, such as chemotherapy and radiation [70,71,72]. PD-L1 expression in tumor is constitutively upregulated by oncogene alterations, such as *PD-L1* and *JAK2* genomic amplification or *PI3K/AKT* pathway activation, and is induced by the interferon gamma (IFN-γ) generated by infiltrating lymphocytes [73,74,75,76]. Therefore, it is difficult to evaluate accurate PD-L1 expression levels at a specific site due to dynamic changes. Secondly, IHC evaluation of PD-L1 expression can be highly variable due to variability in PD-L1 assays and cutoff values for PD-L1 positive expression.

Recently, the Blueprint phase 2 project was conducted to compare five IHC assays—22C3, 28-8, SP142, SP263, and 73-10—for evaluating PD-L1 expression [77]. The 22C3, 28-8, and SP263 assays demonstrated comparable staining results. The SP142 assay exhibited fewer stained tumor cells, while the 73-10 assay exhibited a higher sensitivity than other assays. Based on these results, PD-L1 assay may be indicated as a complementary assay, not as a companion.

### 2.2. Tumor Mutational Burden, Mismatch Repair Deficiency, and Neoantigens

T cell activation requires the tumor antigenic peptide/major histocompatibility complex to interact with the T-cell receptor (TCR), and a costimulatory signal between APCs and T cells [17]. T cell immune response is closely associated with an increased level of immunogenic antigens [78].

Tumor mutation burden (TMB), defined as the total number of nonsynonymous somatic mutations present in a tumor cell, is another potential predictive biomarker for ICIs. However, its clinical utility has yet to be established due to high costs, a long turn-around time, and the limited availability of fresh, unfixed tissue [79]. TMB is generally evaluated by whole-exome sequencing or comprehensive genomic profiling. Nonsynonymous mutations in tumor cells have the potential to generate neoantigens that are recognized by the host immune system, and can lead to a more robust antitumor immune response [80]. Several recent studies have showed promising results for TMB in predicting response to ICI. High TMB has been shown to be associated with improved OS in patients with advanced melanoma treated with ipilimumab, a humanized monoclonal antibody that blocks CTLA-4 [26].

In a phase III trial of first-line therapy with nivolumab for advanced NSCLC, the patients with a high TMB were found to have significantly improved PFS in the nivolumab group compared to the chemotherapy group (9.7 vs. 5.8 months, respectively; hazard ratio (HR) = 0.62; 95% confidence interval (CI), 0.38–1.00) [27]. ORR was 47% for the nivolumab group and 28% for the chemotherapy group. In addition, patients with high TMB and PD-L1 expression of ≥50% had a higher response rate (75%) than those with only one or neither of these factors. Similarly, in a phase III trial of combination therapy with nivolumab and ipilimumab in NSCLC, a TMB of at least 10 mutations per megabase was correlated with longer PFS in the combination group, regardless of PD-L1 expression [81]. Therefore, a high TMB can be a predictive biomarker for immunotherapy.

Increased TMB is also associated with DNA mismatch repair (MMR), which is a DNA repair mechanism of restoring DNA integrity after the occurrence of mismatch errors during DNA replication [82]. Microsatellites, which are repetitive DNA sequences with a unit length ranging from two to five base pairs distributed along coding and noncoding regions of the genome, are particularly prone to mismatch errors due to MMR deficiency (dMMR) [83], and can result in the microsatellite instability-high (MSI-H) phenotype. MSI-H correlates with an increased neoantigen burden, which might sensitize tumors to immunotherapies.

In a phase II study of pembrolizumab for advanced cancers with or without dMMR, the immune-related ORR and immune-related PFS rate at 20 weeks was 40% and 78%, respectively, in the cohort of dMMR CRC as opposed to 0% and 11%, respectively, in the cohort of proficient MMR (pMMR) CRC [28]. In this study, whole-exome sequencing analysis demonstrated a mean of 1782 somatic mutations per dMMR tumor as opposed to 73 mutations per pMMR tumor (*p* = 0.007), and high somatic mutation loads were associated with prolonged PFS (*p* = 0.02). In addition to this study, a total of 149 patients across 15 tumor types with MSI-H or dMMR cancers were enrolled in five single-arm multicohort multicenter trials (KEYNOTE-016, -164, -012, -028, and -158) [29,30,31,32,33]. The ORR for all patients was 39.6% (95% CI, 32–48%), including 7.4% CR and 32.2% partial response, with more than 78% of the patients responding after six months.

These results have led the FDA to approve the use of pembrolizumab in adult and pediatric patients with advanced solid tumors with positive MSI-H or dMMR, regardless of the tumor site or histology. Of note, this is the first tissue-agnostic approval for pembrolizumab by FDA.

In recent years, advances in next-generation sequencing (NGS) and bioinformatics have identified gene mutations specific to cancer cells, enabling the comprehensive analysis of neoantigen candidates resulting from mutations, and the development of personalized cancer vaccine therapies targeting neoantigens. Conventional cancer vaccines are designed to target TAAs, which are overexpressed in cancers. However, because TAAs are also expressed in normal tissues, cancer vaccine against TAAs can potentially initiate central and peripheral tolerance responses, which can result in low vaccination efficiency [84]. Compared to non-mutated self-antigen, cancer neoantigens derived from somatic mutations in the tumor tissue can be recognized as non-self by the immune system, which can make them highly immunogenic and reduce central thymic tolerance [11,85].

The identification of cancer neoantigens for individual patients is the first step in generating personalized vaccines. In general, non-synonymous somatic mutations in cancers, such as point mutations, insertions, or deletions, can be introduced based on whole-exome sequencing analysis and RNA sequencing by comparing tumor and matched normal tissues, such as peripheral blood mononuclear cells [86,87]. Next, coding mutations are predicted based on human leukocyte antigen (HLA) peptide binding affinity analysis to determine the most immunogenic antigen candidates for personalized cancer vaccines [88,89,90]. Although various predictive algorithms for HLA neoantigen binding are available, the reliability of these predictive algorithms is uncertain [88,91,92,93]. The binding affinity of all six HLA class I molecules in humans cannot be predicted yet [85]. A relatively high false positive rate of predicted epitopes is obtained because these methods are derived based on binding affinity data, and therefore only model the single event of peptide‒HLA binding [94]. In addition, most of these methods do not account for the endogenous processing and transport of peptides prior to HLA binding [95]. Furthermore, these methods are less reliable for predicting HLA-II restricted antigenic peptides due to the structural complexity of the endosomal HLA-II peptide [96,97]. Therefore, HLA-II prediction tools will be required to be developed in the future.

Several studies have demonstrated the safety and efficacy of personalized cancer vaccines, using current HLA binding prediction algorithms. In a first-in-human clinical study of advanced melanoma patients treated with an mRNA-based neoantigen vaccine, responses were elicited against 60% of the predicted neoepitopes, and each patient developed T cell responses against at least three mutations [98]. The authors showed the infiltration of neoantigen-specific T cells into the tumor. Moreover, one patient had a complete response of multiple progressing metastases unresponsive to radiotherapy and anti-CTLA-4 antibody. In another phase I study with a personalized neoantigen vaccine in melanoma patients, six patients were vaccinated with synthetic long peptides that target up to 20 predicted personal tumor neoantigens [99]. This vaccine induced polyfunctional CD4+ and CD8+ T cells targeting 58 (60%) and 15 (16%), respectively, of the 97 unique neoantigens used across patients. In addition, four out of six melanoma patients treated with the neoantigen vaccine showed no recurrence at 25 months after vaccination. The other two recurrent patients were then treated with the anti-PD-1 antibody and showed complete tumor regression. A recent phase I clinical trial of glioblastoma treated with a personalized neoantigen long peptide vaccine induced circulating neoantigen-specific CD8+ and CD4+ T cells [86]. This study also showed an increased number of T cells within the intracranial tumor; they are specific to neoantigens and can be targeted by vaccination. Single-cell TCR analysis showed that circulating neoantigen-specific T cells can migrate into an intracranial tumor. Therefore, a neoantigen-based cancer vaccine, which can induce tumor-specific CTLs, should be developed. Several neoantigen cancer vaccines are currently in clinical trials (Table 2). Neoantigens are suggested to provide tumor-specific targets for personalized cancer vaccines.

### 2.3. Tumor-Infiltrating Lymphocytes (TILs)

TILs are polymorphic in nature and are predominantly found in the tumor microenvironment with CD4+, CD8+ T cells, B-cells, and NK cells [100]. The presence of a high number of TILs is consistent with its role in adaptive antitumor immunity for the prevention of tumor progression [34]. The presence of tumor-infiltrating CD8+ T cells, which serve as cytotoxic agents, in the tumor tissue is associated with a better prognosis in melanoma, CRC, and several other cancers [35,36,37]. In a meta-analysis study of NSCLC, the high presence of CD8+ TIL was correlated with improved OS (HR = 0.91; 95% CI, 0.84–0.98, *p* = 0.013), and recurrence or disease-free survival (HR = 0.74; 95% CI, 0.61–0.89, *p* = 0.001) [38].

Conversely, the efficacy of tumor-infiltrating CD4+ T cells may be controversial, which might be explained by the different CD4+ subtypes. CD4+ Th1 and Th2 cells exert cytotoxic effects on tumor cells by secreting IL-2, IL-4, and IFN-*γ* [101]. Regulatory T cells (Tregs), expressing FoxP3, are associated with poor survival as they can suppress tumor-specific T cell immune responses [102,103]. Therefore, the clinical significance according to each subtype remains to be established.

TILs have been investigated as a predictor of response to ICIs. In advanced melanoma patients treated with pembrolizumab, pre-treatment samples from responders showed higher CD8+ T cell densities at both the invasive margin and tumor center than those from non-responders [39]. This study also demonstrated that an increase in the intratumoral CD8+ T cell density from baseline to post-dosing biopsy was significantly associated with a radiographic reduction in tumor size. These results indicate that preexisting immune responses enhanced treatment efficacy by immunotherapy.

Recently, we have reported that the Immunoscore, which quantifies CD3+ and CD8+ expression within the tumor and its invasive margin, provides a reliable estimate of the risk of recurrence in patients with CRC [104]. This result may support the realization of the consensus Immunoscore as a prognostic marker of patients’ responses to immunotherapies.

### 2.4. Tumor-Infiltrating Tregs and Myeloid Cells

The immunosuppressive tumor environment surrounding the tumor is considered to be one of the major barriers to successful cancer immunotherapy. Tregs and myeloid-derived suppressor cells (MDSCs) are major components of the immunosuppressive environment. Hence, these cells can be a promising target in cancer immunotherapy.

Tumor-infiltrating CD4+ Tregs, characterized by CD25 (IL-2 receptor *α* chain) and the master regulatory transcription factor FoxP3 expression, suppress effective antitumor immune response and contribute to tumor progression through IL-10 and transforming growth factor (TGF)-*β*-induced cell-cell contact interaction [105,106,107]. IHC analysis showed that Tregs expression among TILs is associated with poor prognosis in breast cancer patients [108]. A meta-analysis study of 76 articles revealed that high infiltration of FoxP3+ Tregs is associated with poor OS in the majority of cancers, including cervical, renal, melanoma, and breast [40]. Therefore, immunotherapies targeting Treg infiltration can promote the development of an antitumor response. Because CTLA-4 and PD-1 molecules are also expressed by Tregs, ICIs would be more effective for Tregs [41]. Anti-CTLA-4 therapy in bladder cancer patients has been demonstrated to decrease Tregs accumulation in the tumor tissue and increase inducible costimulator (ICOS) + CD4+ T cells, which produce IFN-*γ* and can recognize TAA [41]. Several clinical studies have been conducted on the targeting of intratumoral Treg with anti-CD25, CCR4, OX40, and GITR [42].

MDSCs play a critical role in immunosuppression of the tumor environment by expressing arginase-1, inducible nitric oxide synthase, indoleamine 2,3-dioxygenase, and reactive oxygen species [43,44,45,46]. However, the evaluation of MDSCs might be difficult due to heterogenous populations at different stages of differentiation [109]. Regorafenib has been reported to target a wide range of angiogenic factors (VEGFR1-3, PDGFR-b, and TIE2), oncogenic kinases (c-KIT, RET, FGFR, Raf-1, and BRAF), and the tumor microenvironment [110,111]. Recently, the REGONIVO trial, regorafenib plus nivolumab in advanced gastric cancer or CRC patients, was conducted to test whether regorafenib affects the function of tumor-associated macrophages.

## 3. Biomarkers in Peripheral Blood

Tumor tissue biopsy or surgical removal is invasive and tumor tissues might be difficult to access due to the tumor’s anatomical site. Sometimes, the amount of tissue is not sufficient for evaluation studies. Peripheral blood markers can be used as predictive biomarkers for cancer immunotherapy because they are convenient to assay and are non-invasive. It is also possible to evaluate the dynamic immune reactivity over time.

### 3.1. Lymphocytes and Neutrophils

High lymphocyte and low neutrophil counts were associated with a good prognosis in cancer patients [112,113,114]. In patients with advanced melanoma treated with nivolumab, the absolute lymphocyte count of ≥1000 µ/L (at week 3: HR = 0.40, *p* = 0.004, and at week 6: HR = 0.33 *p* = 0.001) and absolute neutrophil count of ≤4000 µ/L (at week 3: HR = 0.46, *p* = 0.014, and at week 6: HR = 0.51 *p* = 0.046) was significantly associated with OS [114].

The neutrophil-to-lymphocyte ratio (NLR) has been suggested as a simple index of the systemic inflammatory response [47]. High NLR has been shown to be associated with poor response to immunotherapy in advanced cancers [48,49,50]. We have reported that an NLR <3.0 correlates with longer survival in advanced CRC patients receiving the peptide vaccine therapy. Moreover, patients with a lymphocyte count less than 15% should be excluded from immunotherapy [7]. A recent meta-analysis study showed that high pretreatment NLR is associated with significantly poor OS (HR = 1.98, *p* < 0.001) and PFS (HR = 1.78, *p* < 0.001) in 4647 advanced cancer patients treated with immunotherapies such as anti-VEGF/VEGFR and anti-CTLA-4 antibodies [115]. However, the cutoff values for NLR have not been defined yet; several studies have reported different cutoff values to evaluate the prognostic value [7,49,50,115].

The derived NLR (dNLR), calculated as the absolute neutrophil count/(white blood cell count-absolute neutrophil count), may be more relevant than the NLR because it includes monocytes and other granulocyte subpopulations [116]. Recent studies have shown that high dNLR is associated with poor prognosis in several tumors [117,118,119]. In a large cohort of 720 melanoma patients treated with ipilimumab, the patients with dNLR <3 showed improved median OS (9.2 vs. 2.7 months, respectively, *p* < 0.0001) and PFS (4.3 vs. 2.4 months, respectively, *p* < 0.0001) than those with dNLR ≥3 [119]. Moreover, the patients with baseline dNLR ≥3 showed double-increased risks of death (HR = 2.29; 95% CI, 1.86–2.82, *p* < 0.0001) and disease progression (HR = 2.03; 95% CI, 1.66–2.47, *p* < 0.0001) compared to those with lower dNLR. These results indicate that pretreatment with dNLR can be a promising predictive and prognostic biomarker in advanced cancer patients treated with immunotherapy.

### 3.2. Lactate Dehydrogenase (LDH)

LDH is the final enzyme in the glycolysis pathway, and catalyzes the interconversion of pyruvate and lactate [120]. High levels of LDH have been reported in cancer patients owing to the Warburg effect, characterized by increased utilization of glycolysis rather than oxidative phosphorylation for their energy requirement in a well-oxygenized environment [121]. LDH has been established as a negative prognostic factor in advanced melanoma, and high levels of LDH have been recognized as Stage IV melanoma by the American Joint Committee on Cancer Classification [122].

In 230 patients with advanced melanoma treated with ipilimumab in the Netherlands (NL) and the United Kingdom (UK), the median OS for baseline LDH <2.0-fold higher than the upper limit of normal (ULN) and baseline LDH >2.0-fold higher than the ULN were 10.0 and 2.9 months, respectively, (*p* < 0.0001) [51]. Similarly, the median OS for the LDH-low and LDH-high groups in the UK cohort were 5.0 and 3.2 months, respectively (*p* = 0.004). Other studies have confirmed the association between elevated baseline LDH and poor OS in melanoma patients treated with other ICIs, such as pembrolizumab and nivolumab [52,53,54]. A recent meta-analysis of 14 studies involving 4084 patients with lung cancer was conducted to evaluate the prognostic value of pretreatment LDH levels for lung cancer [55]. This study demonstrated that patients with high LDH level at pretreatment exhibit higher risk for death than those with normal LDH levels (HR = 1.49, 95% CI, 1.38–1.59). 

Elevated LDH levels also correlate with short OS in lung cancer patients treated with nivolumab or pembrolizumab [56,57,58]. In a multivariable analysis conducted in advanced esophageal squamous cell carcinoma patients treated with camrelizumab—a humanized anti-PD-1 antibody—elevated baseline LDH level was associated with poor OS [59].

### 3.3. C-Reactive Protein (CRP)

The CRP, an inflammatory marker, serves as an index for the immune status of the host and the degree of tumor progression [123]. CRP has been shown to induce the expression of acute-phase proteins such as neutrophils and predict poor prognosis in several cancers [124,125,126,127]. 

In 1161 RCC patients who underwent surgery, multivariate analysis revealed that CRP level is an independent prognosticator for cancer-specific survival (HR = 1.007, 95% CI = 1.004–1.009, *p* < 0.001) as well as OS (HR = 1.006, 95% CI = 1.004–1.008, *p* < 0.001) [128]. Patients with intermediate and high CRP level exhibited a 1.67- and 2.48-fold increased risk of dying, respectively, compared to those with low CRP level. In a retrospective study of limited-disease small-cell lung cancer patients undergoing thoracic chemoradiotherapy, patients with a high CRP level showed longer OS than those with lower CRP level (median 20 vs. 14 months, respectively, *p* = 0.025) [129]. In a previous study of 95 advanced melanoma patients treated with ipilimumab, decreasing levels of CRP between baseline and at the end of ipilimumab treatment at week 12 were significantly associated with disease control rate and survival [60]. Therefore, increased CRP from baseline may cause inflammation by tumor development and growth or immune-related adverse event rather than an antitumor response from immunotherapy [63].

### 3.4. MicroRNA (miRNA)

miRNAs are endogenously expressed non-coding RNAs consisting of 18–25 nucleotides that regulate gene expression at the post-transcriptional level [130,131]. Aberrant expression of miRNAs has been reported as a prognostic biomarker for several cancers [132,133,134]. In advanced CRC patients treated with a cancer vaccine, the patients with low expression of miRNA-6826 and -6875 in plasma had better OS than those with high expression of miRNA-6826 and -6875 (*p* = 0.048, and *p* = 0.029, respectively) [61]. These results indicate that high expression of miRNA-6826 and -6875 might be related to the suppression of immune competence and these can serve as potential targets for regulating the effect of immunosuppressive factors. In another study for NSCLC patients treated with nivolumab, a signature of seven circulating miRNAs (miRNA-215-5p, -411-3p, -493-5p, -494-3p, -495-3p, -548j-5p, and -93-3p) was significantly associated with improved OS (*p* = 0.0003) [62].

miRNAs can play a crucial role as oncogenes to induce tumorigenesis or as tumor suppressors to inhibit tumor cell proliferation according to their target mRNAs [135]. Therefore, miRNA-targeted therapies are also suitable for therapeutic applications. MRX34, a liposomal miRNA-34a mimic, was used as the first-in-human miRNA cancer therapy [136].

### 3.5. Circulating Tregs and MDSCs

Tregs and MDSCs can also be detected in the blood; several studies have demonstrated that these cells are negatively correlated with clinical response in advanced cancers [42,44,45,46,102,106]. In our retrospective study of 42 patients with advanced pancreatic cancer treated with the DC vaccine, there were no differences in the percentage of Tregs and MDSCs in the blood between responder and non-responder patients at baseline [6]. After 16 weeks of treatment, the percentage of Tregs and MDSCs in responders was significantly lower than that in non-responders (*p* = 0.0495, and *p* = 0.0430, respectively). Similar results have been reported in other studies; low levels of circulating Tregs and MDSCs in the blood after initial administration of immunotherapy were significantly correlated with better disease control and longer OS than high levels of Tregs and MDSCs [60,137].

Therefore, circulating Tregs and MDSCs both at baseline and after cancer immunotherapy may be useful biomarkers for the antitumor immune response. Commonly used anticancer agents, such as cyclophosphamide and metformin, can regulate the number and function of Tregs, and cyclooxygenase-2 inhibitors and cimetidine can suppress MDSCs [138,139,140,141]. These effects contribute to their antitumor capacities by stimulating anticancer immune responses. 

### 3.6. Ki-67 Expression in PD-1+ CD8+ T Cells

T cell exhaustion is a state of T cell dysfunction in the tumor microenvironment. It has been reported that the expression of inhibitory receptors, such as PD-1, CTLA-4, and T cell immunoglobulin mucin-3 (Tim-3), on exhausted T cells results in reduced proliferation and effector functions in tumors [142,143]. In a phase II trial of a peptide vaccine for advanced pancreatic cancer, high expression levels of PD-1 and Tim-3 on CD8+ T cells were significantly associated with a poor prognosis, indicating that these cells may restrict T cell responses [48]. Therefore, combination immunotherapy with blockade of PD-1/PD-L1 may be useful.

Ki-67, a proliferation marker, has been reported as a surrogate biomarker of effector T cell reinvigoration in patients with advanced melanoma and NSCLC treated with anti-PD-1/PD-L1 therapy [64,65]. After treatment, Ki-67+ PD-1+ CD8+ T cells exhibited low Bcl-2 expression, indicating TCR engagement and effector cell differentiation. Furthermore, these cells expressed the activation markers CD38 and HLA-DR, and the costimulatory molecules CD27, CD28, and ICOS. It was reported that increased number of Ki-67+ PD-1+ CD8+ T cells in the blood within 4 weeks of treatment might predict the response and prognosis of cancer patients treated with anti-PD-1/PD-L1 therapy [64].

## 4. Biomarkers in Other Sources

### Microbiome

The microbiome plays an important role in the maintenance of host metabolism and immune system [144]. Recently, the gut microbiome was reported to be associated with various cancers [145]. *Fusobacterium nucleatum* (*Fn*), an anaerobic oral commensal, has been reported to play an important role in CRC development by upregulating the expression of oncogenic and inflammatory genes [146,147,148,149]. High levels of *Fn* in CRC tissues have been shown to be associated with the late stages of tumors and poor prognosis in the Japanese population [149]. An association has also been reported between gut microbiome and clinical outcomes of cancer immunotherapy. Melanoma patients treated with the anti-PD-1 antibody who showed an improved prognosis exhibited highly diverse and abundant gut Ruminococcaceae/Faecalibacterium [66]. In this study, responders had higher levels of effector T cells in the systemic circulation with a preserved cytokine response to anti-PD-1 therapy than non-responders, who had less diverse and highly abundant Bacteroidales, and showed higher levels of Treg and MDSC.

In addition, NSCLC and RCC patients treated with the anti-PD-1 antibody who showed a better response had overabundant *Akkermansia muciniphila* in their stools compared to nonresponders [67]. Moreover, transplantation of *Akkermansia muciniphila* into mice improved the therapeutic efficacy of the anti-PD-1 antibody in an IL-12-dependent manner by increasing the recruitment of CCR9+CXCR3+CD4+ T cell to the tumor site. Although the gut microbiome might be used as a diagnostic and therapeutic biomarker, more confirmatory results are required.

## 5. Conclusions

Cancer immunotherapy using immune checkpoint inhibitors is an emerging strategy in advanced cancers. Identifying and developing biomarkers that can predict responses to cancer immunotherapy and prognosis are important and attractive (Figure 1). Although numerous factors from the tumor tissue, peripheral blood, and other sites might affect response to cancer immunotherapy, their predictive and prognostic abilities have not yet been validated by prospective and randomized clinical trials. Moreover, it would be difficult to predict responses using a single biomarker, due to the complexity of the tumor and human immune system. Therefore, a comprehensive assessment of tumor immunity as a dynamic spatiotemporal process is crucial for effective cancer immunotherapy.

Recently, the new concept of a “cancer immunogram” was proposed to describe the interaction between cancer and the immune system [78]. This framework of the immunogram, consisting of seven parameters, obtained by tumor genomics, immune gene signatures, immunohistochemistry, and a blood-based assay, can be used to identify the most prominent biomarkers in individual cases. Multiparameter biomarkers such as the “cancer immunogram” can be used as an integrated biomarker, and may lead to personalized cancer immunotherapy in the future.

## Figures and Tables

**Figure 1 cancers-11-01223-f001:**
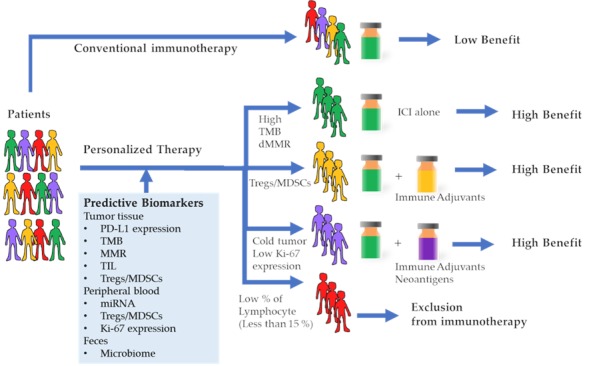
Cancer immunotherapy treatment outcomes based on predictive biomarkers. Conventional immunotherapies were performed without a predictive biomarker, hence the benefit was low. Patients with a high tumor mutation burden (TMB) or deficient mismatch repair gene (MMR) will respond well to the immune check point inhibitor (ICI) alone. Patients with a high amount of regulatory T cells (Treg) and/or myeloid-derived suppressor cells (MDSC) will require combination immunotherapy of ICI and agents that resolve suppressive immunity. Patients with cold tumors and/or low Ki-67 expression in peripheral blood mononuclear cells (PBMC) will require a combination therapy of ICI and immune adjuvants as well as vaccination against the neoantigens. Patients with a low percentage of peripheral lymphocytes might be regarded as unsuitable candidates for immunotherapy.

**Table 1 cancers-11-01223-t001:** Biomarkers for cancer immunotherapy.

Sample	Biomarker	Clinical Significance
Tumor Tissue	PD-L1	Increased expression in tumor cells is associated with a positive clinical response to anti-PD-1/PD-L1 antibodies [16,21,22,23,24,25].
	TMB	High TMB is associated with an improved response to ICIs [26,27].
	MMR	MMR deficiency, regardless of tumor types, correlates with a clinical response to pembrolizumab [28,29,30,31,32,33].
	TIL	The presence of tumor-infiltrating CD8+ T cells is associated with a better prognosis [34,35,36,37,38,39].
	Tregs/ MDSCs	The presence of tumor-infiltrating Tregs and MDSCs is associated with a poor prognosis [40,41,42,43,44,45,46].
Peripheral Blood	Neutrophils/leukocytes	Elevated NLR is associated with a poor response [7,47,48,49,50].
	Low % of lymphocytes	Patients with a lymphocyte count less than 15% should be excluded from immunotherapy [7].
	LDH	Elevated pretreatment levels are correlated with a worse OS [51,52,53,54,55,56,57,58,59].
	CRP	High level of CRP is associated with an increased risk of OS [60].
	miRNA	Expression of several miRNAs is associated with clinical outcomes [61,62].
	Tregs/ MDSCs	Decreased level of circulating Tregs and MDSCs after treatment correlated with improved prognosis [6,60,63].
	Ki-67 expression	Increased Ki-67 expression in PD1+ CD8+ T cells is associated with a good response [64,65].
Feces	Microbiome	The presence of Ruminococcaceae/Faecalibacterium in melanoma patients and *Akkermansia muciniphila* in NSCLC and RCC is associated with a clinical response to anti-PD-1 antibody [66,67].

PD-L1, programmed cell death-1 ligand 1; PD-1, programmed cell death receptor-1; TMB, tumor mutation burden; ICIs, immune checkpoint inhibitors; MMR, mismatch repair; TIL, tumor-infiltrating lymphocyte; Treg, regulatory T cells; MDSC, myeloid-derived suppressor cells; NLR, neutrophil-to-lymphocyte ratio; LDH, lactate dehydrogenase; OS, overall survival; CRP, C-reactive protein; miRNA, micro ribonucleic acid; NSCLC, non-small-cell lung carcinoma; RCC, renal cell cancer.

**Table 2 cancers-11-01223-t002:** Ongoing clinical trials with neoantigen cancer vaccines.

Trial Number	Phase	Disease	Strategy	Sponsor/Investigator
NCT03639714	1/2	NSCLC, CRC, Gastroesophageal cancer, Urothelial cancer	Adenovirus vector	Gritstone Oncology, Inc.
NCT03313778	1	Solid tumors	mRNA	ModernaTX, Inc.
NCT03871205	1	Lung cancer	DC	Shenzhen People’s Hospital/Lili Ren
NCT03674073	1	HCC	DC	Chinese PLA General Hospital/Ping Liang
NCT02956551	1	NSCLC	DC	Sichuan University/Zhen-YU Ding
NCT03532217	1	Prostate cancer	DNA	Washington University School of Medicine/Russell Pachynski
NCT03122106	1	Pancreatic cancer	DNA	Washington University School of Medicine/Gillanders et al.
NCT03199040	1	Breast cancer	DNA	Washington University School of Medicine/William Gillanders
NCT04015700	1	Glioblastoma	DNA	Washington University School of Medicine/Gavin Dunn
NCT03988283	1	Pediatric recurrent brain tumor	DNA	Washington University School of Medicine/Karen M Gauvain
NCT03645148	1	Pancreatic cancer	Peptide	Zhejiang Provincial People’s Hospital
NCT03558945	1	Pancreatic cancer	Peptide	Changhai Hospital/Gang Jin
NCT03662815	1	Advanced solid tumor	Peptide	Sir Run Run Shaw Hospital
NCT03715985	1	Melanoma, NSCLC, Renal cell carcinoma	Peptide	Herlev Hospital/Inge Marie Svane
NCT02950766	1	Renal cell carcinoma	Peptide	Patrick Ott
NCT03422094	1	Glioblastoma	Peptide	Washington University School of Medicine/Gavin Dunn
NCT03361852	1	Follicular lymphoma	Peptide	Dana-Farber Cancer Institute/Eric Jacobsen
NCT03929029	1	Melanoma	Peptide	Dana-Farber Cancer Institute/Patrick A Ott
NCT03068832	1	Pediatric brain tumor	Peptide	Washington University School of Medicine/Karen M Gauvain
NCT03606967	2	Breast cancer	Peptide	National Cancer Institute/William Gillanders
NCT03219450	1	Lymphocytic leukemia	Peptide	Dana-Farber Cancer Institute/Pavan Bachireddy

NSCLC, non-small-cell lung carcinoma; CRC, colorectal cancer; mRNA, messenger ribonucleic acid; DC, dendritic cell; HCC, hepatocellular carcinoma; DNA, deoxyribonucleic acid.

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
