# Peer review of "Novel Biomarkers for Personalized Cancer Immunotherapy"

_cancers, 2019, doi:10.3390/cancers11091223_

Round 1
Reviewer 1 Report
This is a very well written review article which uses a wide range of published literature to describe how biomarkers may potentially be used to direct cancer immunotherapy.
The text is comprehensive, very well written, and the use of English is excellent.
I have no major concerns regarding organization of the text.
- A minor point is that the introduction of figure one may be more appropriate towards the end of the article when the different themes on the chart have been discussed in the text.
- In addition, where the text talks about the use of CRP as a biomarker, I think it would be sensible to re-word the phrase that says “2.48 increased risk for overall survival”. The conjunction of ‘risk’ and improved survival seems awkward.
- I wonder if the authors might comment on the relative significance of PD-L1 expression on tumor cells or infiltrating reactive lymphocytes ? This has been investigated in previous studies and is of interest in relation to how PD-1 blockade may work.
Author Response
Response to Reviewer 1 Comments
We wish to express our appreciation to the Reviewer for insightful comments, which have helped us significantly improved the paper.
Point 1: A minor point is that the introduction of figure one may be more appropriate towards the end of the article when the different themes on the chart have been discussed in the text.
Response 1: Thank you for your valuable advice. As the reviewer points out, we have inserted the figure 1 in Conclusion section.
Point 2: In addition, where the text talks about the use of CRP as a biomarker, I think it would be sensible to re-word the phrase that says “2.48 increased risk for overall survival”. The conjunction of ‘risk’ and improved survival seems awkward.
Response 2: As the reviewer points out, we have corrected the phrase as follows (Page 9, line 323). “ 2.48-fold increased risk of dying”.
Point 3: I wonder if the authors might comment on the relative significance of PD-L1 expression on tumor cells or infiltrating reactive lymphocytes ? This has been investigated in previous studies and is of interest in relation to how PD-1 blockade may work.
Response 3: Thank you for your important comments. We have added comment about PD-1/PD-L1 interaction and treatment strategy of PD-1 blockade (Page 3, line 74).
Thank you again for your comments on our paper. We trust that the revised manuscript is suitable for publication.
Reviewer 2 Report
I think it is a nice overview of which biomarkers might be used for cancer immunotherapy. In the conclusions the authors suggest to use the concept of immunogram to use as an integrated biomarker! This is of course an interesting option. However, in the manuscript a number of biomarkers are described and all have their advantages and limitations. It would have been nice if the authors themselves could come with suggestions on a combination of biomarkers described in the manuscript which might be used. Now it is just a mere description of all kind of different biomarkers.
It is certainly not easy to make suggestions as e.g. in patients with negative the PD-L1 expression in tumors also seem to have clinical benefit from anti PD1/PD-L1 treatment (page 3 line 89). Will this come out with the help of an immunogram?
The same for markers that are expressed in normal tissues. Will an immunogram help with those markers?
So it would be nice if in the concluding remarks the authors could summarize some suggestions of combinations of biomarkers described in the manuscript which can be used for prediction or prognosis.
Author Response
Response to Reviewer 2 Comments
We wish to express our strong appreciation to the Reviewer for insightful comments on our paper. We feel the comments have helped us significantly improve the paper.
Point 1: It is certainly not easy to make suggestions as e.g. in patients with negative the PD-L1 expression in tumors also seem to have clinical benefit from anti PD1/PD-L1 treatment (page 3 line 89). Will this come out with the help of an immunogram?
Response 1: The reviewer asks a very interesting question. We suggested the limitation of PD-L1 expression as a biomarker, due to the tumor heterogeneity, the dynamic expression evolving after therapies, variability in PD-L1 assays and cut-off values for PD-L1 positive expression in the manuscript (Page 4, line 104-112). As the reviewer points out, it might be associated with other parameters such as TMB, consisting of the immunogram. For example, we mentioned that a TMB of at least 10 mutations per megabase was correlated with longer PFS in the combination group, regardless of PD-L1 expression (Page4, line139). Therefore, we have added the sentence as follows (Page 4, line 101). “In addition, other predictive biomarkers of response to anti-PD-1/PD-L1 treatment might be involved.”
Point 2: The same for markers that are expressed in normal tissues. Will an immunogram help with those markers?
Response 2: Thank you for your important comments. The immunogram is consisted of seven unrelated immunogenic parameters: tumor foreignness (mutational load), general immune status (lymphocyte count), immune cell infiltration capacity (intratumoral T cells), absence of checkpoints (PD-L1 expression on tumor cells), absence of soluble inhibitors (IL-6, CRP), absence of inhibitory tumor metabolism (LDH), and tumor sensitivity to immune effector mechanisms (MHC expression). Therefore, the immunogram might not be associated with the markers that are expressed in normal tissues. Hopefully this answers the reviewer’s question.
Point 3: So it would be nice if in the concluding remarks the authors could summarize some suggestions of combinations of biomarkers described in the manuscript which can be used for prediction or prognosis.
Response 3: We thank the reviewer for this valuable comment. We agree with you. Unfortunately, we have no data and were not able to summarize some suggestions of combinations of biomarkers in the manuscript. Therefore, we suggested to use the concept of cancer immunogram to use as an integrated biomarker. In the future, we would like to identify multiple and multiplexed personalized biomarkers.
Thank you again for your comments on our paper. We trust that the revised manuscript is suitable for publication.
Reviewer 3 Report
Page 3, line 74/75: please give a short explanation about PD-1 / PD-L1 interaction
Page 4, line 120: please provide a short explanation about the difficulties for clinical establishment of TMB, like high costs, dispute about cut-offs, panel sizes etc.
Page 8, line 275/276: please give references for the mentioned "several studies"
Author Response
Response to Reviewer 3 Comments
We wish to express our appreciation to the reviewer for insightful comments on our paper. The comments have helped us significantly improve the paper.
Point 1: Page 3, line 74/75: please give a short explanation about PD-1 / PD-L1 interaction.
Response 1: Thank you for your valuable advice. We have added a short explanation about PD-1/PD-L1 interaction (Page 3, line 74-78).
Point 2: Page 4, line 120: please provide a short explanation about the difficulties for clinical establishment of TMB, like high costs, dispute about cut-offs, panel sizes etc.
Response 2: Thank you for your important comments. As the reviewer points out, we have added a short explanation and reference (Page 4, line 124-125).
Point 3: Page 8, line 275/276: please give references for the mentioned "several studies"
Response 3: In accordance with the reviewer’s comment, we have added the references (Page8, line281).
We wish to thank the Reviewer again for your valuable comments.